# Does the Environmental Air Impact the Condition of the Vomeronasal Organ? A Mouse Model for Intensive Farming

**DOI:** 10.3390/ani13121902

**Published:** 2023-06-07

**Authors:** Violaine Mechin, Patrick Pageat, Marion Boutry, Eva Teruel, Céline Portalier, Pietro Asproni

**Affiliations:** 1Tissular Biology and Chemical Communication Department, IRSEA, Institute of Research in Semiochemistry and Applied Ethology, 84400 Apt, Francep.asproni@group-irsea.com (P.A.); 2Research and Education Board, IRSEA, Institute of Research in Semiochemisrty and Applied Ethology, 84400 Apt, France; p.pageat@group-irsea.com; 3Statistics and Data Management Service, IRSEA, Institute of Research in Semiochemisrty and Applied Ethology, 84400 Apt, France; e.teruel@group-irsea.com; 4Animal Experimentation Department, IRSEA, Institute of Research in Semiochemistry and Applied Ethology, 84400 Apt, France

**Keywords:** vomeronasal organ, environmental air, alteration, ammonia, histology

## Abstract

**Simple Summary:**

The vomeronasal organ plays an essential role in animal life, ensuring chemical communication and thus maintaining harmony in social groups. Induced and spontaneous alterations of this organ have been linked to the onset of behavioral disorders in animals. To date, the environmental condition has not been investigated as a possible cause of vomeronasal organ inflammation, even though the pollutant gas emitted by litter decomposition on farms has proven to be responsible for respiratory tract inflammation. The aim of this study was to analyze the impact of the farming environment on the vomeronasal organ condition. The results showed that this organ is significantly impacted by the confined environment, providing new reasons to improve farming conditions.

**Abstract:**

Chemical communication in mammals is ensured by exchanging chemical signals through the vomeronasal organ (VNO) and its ability to detect pheromones. The alteration of this organ has been proven to impact animal life, participating in the onset of aggressive behaviors in social groups. To date, few studies have highlighted the possible causes leading to these alterations, and the farming environment has not been investigated, even though irritant substances such as ammonia are known to induce serious damage in the respiratory tract. The goal of this study was to investigate the environmental impact on the VNO structure. Thirty mice were split into three groups, one housed in normal laboratory conditions and the other two in confined environments, with or without the release of litter ammonia. VNOs were analyzed using histology and immunohistochemistry to evaluate the effect of different environments on their condition. Both restricted conditions induced VNO alterations (*p* = 0.0311), soft-tissue alteration (*p* = 0.0480), and nonsensory epithelium inflammation (*p* = 0.0024). There was glycogen accumulation (*p* < 0.0001), the olfactory marker protein was underexpressed (*p* < 0.0001), and Gαi2 positivity remained unchanged while Gαo expression was upregulated in confined conditions. VNO conditions seemed to worsen with ammonia, even if not always significantly. These murine model results suggest that the housing environment can strongly impact VNO conditions, providing novel insights for improving indoor farming systems.

## 1. Introduction

The vomeronasal organ (VNO) plays a crucial role in animal communication since it is responsible for semi-chemical reception in most animals. It is composed of a sensorial (VNSE) and a nonsensory (NSE) epithelium arranged around a lumen where the molecules are transported [1,2].

In mice, the VNO is a well-developed sensory organ composed of vomeronasal type 1 receptors (V1R), type 2 receptors (V2R), and formyl peptide receptors (FPRs) neurons [3,4]. V1Rs and V2Rs cells are responsible for the reception of chemical cues and transmitting these signals to the accessory olfactory bulb (AOB), inducing innate behaviors in response to these stimuli [1,2,5]. These vomeronasal neurons are identifiable through the presence of Gαi2 and Gαo proteins, respectively, in the VNO neuron cytoplasm [4,6]. These properties give this organ a crucial role in chemical communication and, thus, in behavior and social harmony in groups.

Many studies have shown that experimental alterations or ablations of these organ-induced social difficulties in maternal recognition or mating or increased aggression in social groups [7,8,9,10,11]. The occurrence of spontaneous alterations has rarely been investigated, but recent studies have highlighted the link between vomeronasalitis and aggressive behaviors in cats and pigs [12,13]. To date, the causes of these alterations have never been investigated.

The olfactory epithelium is strongly impacted by air pollutants in the closed housing environment of many farm species, such as pigs or rabbits [14,15,16,17]. The organic matter litter decomposition releases some pollutants, such as ammonia [18,19,20], responsible for nasal pathologies and contributes to the deterioration and inflammatory lesions of the nasal epithelium in mice [21,22]. Since the vomeronasal epithelium shares several anatomical and physiological similarities with the olfactory mucosa, these farm pollutants could also be presumed to be possible causes of inflammation of the VNO. A study by Gaafar showed that prolonged exposure to ammonia had a chronic irritative effect on the respiratory mucosa and the VNSE with a reduction in the number of neurons in rabbits [16].

Therefore, the aim of this study was to evaluate the effect of different environmental conditions on the vomeronasal organ state using a mouse model. VNO cellular changes were investigated in mice housed in normal conditions and compared to those maintained in a plexiglass box in the presence or absence of natural ammonia, which is emitted by litter decomposition. Morphological changes in the VNO were assessed using histological analyses to evaluate possible VNO alterations and inflammation. The presence of accumulated glycogen was also evaluated since it is a relevant indicator of metabolic alteration, and immunohistochemistry was used to analyze the expression of the olfactory marker protein (OMP) as a marker of mature neurons and the expression of Gαi2 and Gαo proteins, which indicated the presence of type 1 or type 2 receptors neurons, respectively.

## 2. Materials and Methods

### 2.1. Animals and Sampling Procedures

The experiment included 30 mice split into 3 groups: 10 mice housed under normal conditions, 10 mice housed under confined conditions without ammonia exposure, and 10 mice housed under the same confined conditions, with natural ammonia exposure. Each group was composed of C57/6Jrj strain mice aged three months, which were provided by Janvier Lab (Le Genest St Isle, France). Five males and five females were used in each group to verify the homogeneity of results concerning sex. Mice were maintained in the IRSEA facility at a temperature of 22 ± 2 °C and 60 ± 20% humidity with a 12 h/12 h light/dark cycle and access to food and water ad libitum. Poplar wood litter was used in each cage (Cat. Aspen; Serlab, Montataire, France).

Dioxygen and ammonia levels were measured daily at 9 a.m. during the experiment with an oximeter (Greisinger, Regenstauf, Germany) and a GasBadge Pro ammonia detector (Industrial Scientific Corporation, Pittsburgh, PA, USA).

For the group in normal laboratory conditions, mice were housed in conventional open-top cages covered by a top grill in a ventilated room of 60 m^3^, shared with 60 other mice involved in other studies of the institute. The litter was changed every two weeks to ensure the ammonia emission in the environmental air stayed at a concentration of 0 ppm. For the two groups in confined conditions, four normal cages containing two or three animals each (ten mice per group) were placed in a plexiglass box of 0.25 m^3^ with two lateral holes (Ø 4 cm), permitting enough dioxygen provision to the housed mice (20 ± 0.5% O_2_) (Figure 1).

For the group not exposed to ammonia, the litter in cages was changed every three days to prevent any ammonia release in the closed environment. The same distribution of animals was used in the third group, and half of the litter was changed every seven days to permit natural ammonia release. Daily levels are presented in Figure 2.

After 21 days of the experiment, all mice were euthanized with an intraperitoneal injection of sodium pentobarbital (Dolethal^®^ 100 mg/kg, Vetoquinol, Lure, France) according to EU animal welfare standards; the mouse heads were collected and placed in a 10% formalin solution (pH 7.4) for one week and decalcified for 48 h in a decalcifying solution (DDK, Milan, Italy). They were transversally cut to obtain sections containing the vomeronasal organ and paraffin-embedded according to routine histological methods. Sections of 3.5 µm thick were obtained in the middle extend of the VNO using a microtome and were mounted on SuperFrostPlus™ slides (Cat No. 10149870, Thermo Fisher Scientific, Illkirch, France). The sections were dried overnight at 37 °C and finally deparaffined with xylene and rehydrated through alcohol, as usual, to proceed to histological and immunohistochemical staining.

### 2.2. Histopathological Analyses

After rehydration, sections were stained with hematoxylin and eosin (HE, BioOptica, Milan, Italy) to perform the histopathological analyses. Each VNO was classified on a scale score according to its general condition, evaluated as 0 (healthy/normal), 1 (weak degeneration signs), or 2 (moderate and strong alteration signs), as demonstrated in Figure 3, considering the presence of aberrant vacuolization or a lower density of the tissue (Figure 3A) in all VNO components (VNSE, NSE, VNO soft tissue). The presence of inflammatory infiltrates was also investigated in the soft tissue only in the NSE (Figure 3B) since soft tissue in the VNSE is absent or negligible in mice. The presence of abnormal amounts of lymphocytes was then classified as 0 (healthy), 1 (few lymphocytes), and 2 (strong lymphocyte infiltration). Finally, the VNSE conditions were assessed using the same scoring system to assess the VNO general condition.

Periodic acid–Schiff (PAS) staining (Cat No. C062AA, DiaPath SpA, Martinengo, Italy) was also performed since it permits the detection of glycogen accumulation in neuronal tissues. Images were analyzed with ImageJ^®^ software (US National Institute of Health, Bethesda, MD, USA) using the “Color Deconvolution” plugin and the “H-PAS” vector. The accumulated glycogen was stained in a characteristic red-purple color in the sensorial epithelium, and positive pixels were analyzed to obtain a percentage of positive staining in the selected surface of the VNSE, as previously described [23].

### 2.3. Immunohistochemical Analyses

To proceed to immunohistochemical (IHC) techniques, slides were put into a microwave with an antigen retrieval solution pH 6 (Cat No. F/T0050; DiaPath SpA, Martinengo, Italy) at 560 W for 3 min and 30 s, followed by 210 W for 15 min. A peroxidase blocking solution (Cat No. ACA 500; Scytek, Logan, UT, USA) was applied and incubated for 30 min at RT in the dark, followed by an UltraVision Protein Block solution (Cat No. TA-125-PBQ, Thermo Scientific, Carlsbad, CA, USA) incubation, for 10 min. After rinsing, sections were incubated for 1 h at room temperature with the following primary antibodies: rabbit anti-Olfactory Protein (OMP) (Cat No. O7889, Sigma, Saint Louis, MO, USA) diluted to 1:10,000, rabbit anti-Gαi2 protein (Cat No. Ab157204, Abcam, Cambridge, UK) diluted 1:200, and goat anti-GαO protein (Cat No. Sc267669, Santa Cruz Biotechnologies, Santa Cruz, CA, USA) diluted 1:200, according to previous studies [23].

After rinsing, a 10 min incubation with a secondary anti-rabbit or anti-goat biotinylated antibody was performed (Cat No. T/ABE125, UltraTek, ScyTek Laboratories, Logan, UT, USA, or Cat No. 12694067, Invitrogen, Carlsbad, CA, USA). The presence of proteins was revealed after the addition of 3,3′ diaminobenzidine tetrahydrochloride (ImmPact^®^ DAB Peroxidase Substrate, Cat No. SK4105, Vector Laboratories, Burlingame, CA, USA). Finally, slides were counterstained with hematoxylin before being dehydrated with gradient series of ethanol, cleared with xylene, and mounted. As a negative control, the primary antibodies were replaced with nonimmune rabbit and goat sera.

Immunolabeled sections were observed with the EVOS^®^ FL Auto Imaging System (Thermo Fisher Scientific, Illkirch, France), and images were captured for analysis. IHC positivity for each protein was obtained with ImageJ^®^ software, the Color Deconvolution plugin, and the “H-DAB” vector and was expressed as the percentage of positive pixels in the selected area, corresponding to the VNSE, as previously reported [23].

### 2.4. Statistical Analysis

Data analysis was performed using R version 4.2.1 (2022-06-23 ucrt) software and RStudio version 1.4.1103 Copyright© (R Foundation for Statistical Computing, Vienna, Austria). The significance threshold was set at 5%.

This study compared three groups of ten mice (normal housing conditions, in the box without ammonia, and in the box in the presence of ammonia). Sex (M/F) was also analyzed to determine whether the effect of the group was different depending on the sex of the mouse. Two VNOs (left and right) were available for each mouse. Thus, the animal was considered a random factor in the models.

For the tissue description and qualitative variables such as the VNO alteration, NSE inflammation, soft tissue, and NSE alteration scores (characterized by a 3-level score: normal, weak, moderate/strong), mixed ordinal logistic regressions (MOLR) were performed. Multiple comparisons that were not available for the multinomial distribution were performed by computing odds ratios.

Concerning the continuous variables (glycogen accumulation, OMP expression, Gαi2, and Gαo proteins), general linear mixed models (GLMMs) were applied. The assumption of homoscedasticity and model residue normality was verified, so the model was carried out using the raw data (OMP expression). When the assumption of normality was violated, a Box–Cox transformation was applied to the data, and GLMM was performed with the transformed data (glycogen accumulation, Gαi2, and Gαo proteins). In all cases, multiple comparisons were performed using the Tukey–Kramer adjustment.

## 3. Results

The following table (Table 1) summarizes the parameters analyzed, the statistical method used for the analysis, and the results of the group effect. Some data are missing due to technical reasons.

### 3.1. VNO Alteration

The statistical analysis revealed a significant difference in VNO condition between the different groups (DF = 2; χ^2^ = 5.95; *p* = 0.0311, MOLR). The differences were found with multiple comparisons and showed that normal housing conditions induced fewer alterations in the mouse VNO than confined conditions without ammonia (*p* = 0.0152) and with ammonia (*p* = 0.0135). No significant differences were detected between the two confined conditions (*p* = 0.2420) (Figure 4).

### 3.2. NSE Inflammation

The data concerning NSE inflammation showed a significant effect (DF = 2; χ^2^ = 12.04; *p* = 0.0024, MOLR), and the results indicated that confined conditions without ammonia increased VNO alterations (*p* = 0.0171) compared to those of mice in normal housing conditions. The same observations were found between NSEs from mice housed in confined conditions in the presence of ammonia (*p* = 0.0114) and mice housed in normal conditions. No significant differences were detected between the effects of the two confined conditions (*p* = 0.251) (Figure 5).

### 3.3. Soft Tissue Alteration

Concerning the alteration of the soft tissue alteration, the statistical analysis showed a significant difference between groups (DF = 2; χ^2^ = 5.91; *p* = 0.0480, MOLR). Multiple comparisons indicated that all groups of mice were significantly different concerning this parameter. The group exposed to ammonia possessed more soft tissue alterations than the other two groups, as well as the non-ammonia group, compared to the normally housed group (*p* = 0.0001 each) (Figure 6).

### 3.4. VNSE Alteration

The statistical analysis showed that the VNSE presented no significant differences in alteration between the mice belonging to the three housing conditions (DF = 2; χ^2^= 2.9825; *p* = 0.2251, MOLR) (Figure 7).

### 3.5. Glycogen Accumulation

A significant difference between the positive PAS staining of the tested groups was found (DF = 2; χ^2^ = 47.93; *p* < 0.0001, GLMM). Multiple comparisons indicated that significantly fewer glycogen deposits were found in the VNSE of mice housed in normal conditions than in the VNSE of mice housed in confined conditions, with or without ammonia (*p* < 0.0001 each). No differences were obtained comparing the presence or absence of ammonia in the plexiglass box (*p* = 0.73) (Figure 8).

### 3.6. OMP Expression in the VNSE

OMP expression was significantly different between groups (DF = 2; χ^2^ = 138.47; *p* < 0.0001; GLMM). The multiple comparisons revealed that the VNSEs of mice housed in normal conditions presented higher OMP expression levels compared to those of both groups housed in confined conditions (*p* < 0.0001 each). No differences were obtained between the two confined housing groups (*p* = 0.6320) (Figure 9).

### 3.7. Gαi2 and Gαo Expression

Statistical analyses showed that Gαi2 expression was not significantly different between groups (DF = 2; χ^2^ = 1.89; *p* = 0.3892, GLMM) (Figure 10).

Gαo protein expression was significantly different between mice in different housing conditions (DF = 2; χ^2^ = 21.462; *p* = 0.0114; GLMM). Post hoc multiple comparisons indicated that the protein expression levels of Gαo were higher in mice housed in confined conditions with or without ammonia than in mice housed in normal conditions (*p* < 0.0001 and *p* = 0.0037, respectively). No differences were observed between the two confined conditions (*p* = 0.4239) (Figure 11).

## 4. Discussion

Vomeronasalitis has been previously associated with aggressive behaviors in farm pigs, indicating that this VNO pathology may contribute to the reduced welfare of animals living in intensive farming conditions [12,13,24]. Considering the particular environmental conditions farm animals are continuously exposed to, the aim of this study was to analyze the effect of these conditions on the mouse VNO to clarify the causes of VNO changes. Our study showed that compared to normal housing conditions, confined conditions induced more VNO changes, suggesting that the accumulation of pollutant gases and dust could be one of the causes of these changes in farm animals. These findings are in agreement with those reported in nasal mucosa studies, and alterations in the nasal mucosa have been previously linked to environmental contaminants such as organic dust and natural gases, including ammonia and hydrogen sulfide [14,15,16]. The olfactory epithelium and the sensory epithelium are physiologically and anatomically similar [25], and it is possible to draw a parallel between these two structures and their responses to environmental conditions.

The NSE analyses showed a significant increase in inflammatory indicators in the VNOs of mice housed in confined conditions compared to those in the VNOs of mice housed in normal conditions. The presence of ammonia appears to increase inflammation, with only 6% of VNOs classified as “normal” or without inflammation in mice exposed to ammonia, compared to the 37% of normal VNOs in mice housed in the same conditions without ammonia exposure, although these data were not significant.

In the soft tissue, a significant increase in alteration signs was revealed in the VNOs of mice housed in confined conditions compared to mice housed in normal conditions. The presence of ammonia significantly exacerbated these changes and indicated that the composition of environmental air had a strong negative impact on this portion of the VNO.

Analysis of the VNSE alteration did not show significant differences between mice in the three housing conditions, but the descriptive data appear to indicate that the presence of ammonia induced more VNSE alterations than those induced by the other housing conditions. Indeed, in mice housed in normal conditions, 45% did not present any signs of VNSE alteration, and only 10% presented moderate or strong signs. In contrast, in the presence of ammonia, only 18% of VNSEs were classified as normal, while 41% presented moderate or strong alterations. One of the possible explications for the lack of significance could be that the ammonia levels were too low to induce significantly different responses in the epithelium. In fact, ammonia was proven to induce strong nasal pathologies on Day 7 when mice were housed in a cage between 100 and 400 ppm [21]. In our study, the average concentration of ammonia exposure was 44 ppm for 21 days. Furthermore, it is well known that the entry of air into the VNO is regulated by pumping/suction mechanisms that make the VNO inaccessible for the majority of the time and, as a consequence, physiologically less exposed to air contaminants than the nasal epithelium [12,16], which reinforces the notion that the ammonia concentration was too low to induce significant modifications in this organ. Additionally, it is possible that the impact of the presence of ammonia would be stronger on farms with poorer environmental conditions. Our data also appear to suggest that ammonia is not the primary cause but contributes to the severity of the alterations triggered by confined conditions. Some studies have investigated pollutant emissions in pig farms, and several other odorant gases have been described in addition to ammonia, such as nitrous oxide (N_2_O), methane (CH_4_), and several phenols, ketones, and aldehydes [26]. It is the authors’ opinion that the simultaneous presence of some of these gases can also contribute to the onset of VNO alterations in farm animals.

The findings presented here are supported by the analysis of the VNO score, which in our study was used to describe the loss of the tissue density of the organ in its totality and its abnormal vacuolization. This alteration could be the result of water or lipid lysosomal accumulation, a known sign of cellular metabolism dysfunction [27]. The mice housed in normal conditions presented fewer VNO changes than those in confined conditions, with or without ammonia. These observations suggest that confined conditions may impact this organ through the accumulation of gases and dust in the environment. Future studies could explore dust levels in normal and confined conditions to identify the effect of dust on the onset of VNO changes since it is well-known that organic dust induces acute lower airway disorders with systemic inflammation in pigs [28]. In fact, it is possible that the effects of ammonia and other gases could be enhanced by excess dust exposure due to insufficient ventilation.

Glycogen is naturally present in nervous tissues and is necessary for metabolism efficiency, as it is a vital energy source for neurons [29,30]. Aberrant accumulation is known to be linked to neuronal degeneration and is frequently found in neurodegenerative diseases such as Parkinson’s disease, Alzheimer’s disease, or dementia [31,32]. Recent studies have highlighted the association between glycogen accumulation and exposure to environmental contaminants in different tissues, such as the lungs [33] or liver [34]. In our study, no aberrant quantities of glycogen were found in the VNSEs of mice housed in normal conditions, but confined conditions, with or without ammonia, seem to induce glycogen accumulation in this organ, suggesting possible cellular metabolism disorder.

To further evaluate the molecular changes induced by confined housing conditions, we analyzed the expression of the olfactory marker protein in vomeronasal mature neurons. This protein is known to have a central role in olfactory nervous system signal transmission [35]. In the present study, a downregulation in OMP expression in confined groups compared to the normal group was observed. Considering that air pollution induces neuronal damage, loss, and neuroinflammation due to mitochondrial dysfunction and oxidative stress in the brain [36], a possible explanation for our observations could be that air contaminants could also have an impact on other neurological organs, such as the VNO, which could induce changes in OMP synthesis, accumulation, and expression. Moreover, we previously showed that OMP levels are also decreased during the aging process in mice due to the degenerative changes induced by this biological phenomenon; thus, this protein may be commonly negatively influenced by exogenous and endogenous insults [23]. This hypothesis needs to be further investigated to better understand these mechanisms.

The Gαi2 protein is known to be expressed on V1Rs neurons in the VNSE and is responsible for the reception of small organic volatile and steroidal molecules involved in social communication, sexual behaviors, and maternal recognition [6,37]. In our study, its expression does not seem to be modified by environmental conditions, as observed in the analysis of the VNSE alteration indicators. However, both parameters need to be evaluated after experiments using stronger concentrations of ammonia, similar to the concentrations used in other studies on the VNSE or other sensorial epithelia, such as the respiratory system, in the presence of this gas [16,21,22].

Type 2 receptors, identified by the immunohistochemical staining of the Gαo protein, detect larger molecules involved in territorial marking or alarm pheromones [38]. The present study indicates that Gαo expression is upregulated in mice housed in a confined environment. Although these data are unexpected, we previously observed a similar finding in the VNO of aging mice [23]. These results can be explained by the ability of V2R to coexpress the M10 family of MHC class 1b molecules linked to the β2microglobuline (β2 m) protein [39]. The levels of these molecules are known to be increased in response to stressful situations, such as age [40], infection, cellular death, or lesions [41], in other sensory epithelia, such as the retina, and in various brain regions. This interpretation must be further investigated and validated with new studies to clarify the observed effect.

In summary, this study allowed us to evaluate the effect of environmental air composition on the VNO condition. The tested parameters revealed a strong and negative impact of a confined environment. These results corroborated those on the negative effect of intensive farming conditions on the respiratory tract due to increasing environmental contaminants due to high animal density [14,42]. Intensive farming and insufficient ventilation lead to high concentrations of ammonia, nitrous oxide, and methane emissions, responsible for severe alterations in the epithelia. Additionally, small dust particles have been proven to deeply enter the respiratory epithelium and induce strong alterations [43,44]. In this study, the presence of ammonia appeared to trigger changes in most of the tested parameters, even if the changes were not always significant. Other studies investigating nasal epithelium found serious damage due to higher ammonia concentrations [21,22,45], and it is the authors’ opinion that the VNSE could also be more severely affected by higher ammonia concentrations. VNO alterations have been proven to be linked to behavioral abnormalities, and VNO inflammation was associated with aggressive behaviors against congeners in pigs [12,13]. These troubles could be explained by a negative impact of epithelium inflammation on the signal transmission efficiency, as previously described in the physiologically similar olfactory epithelium, which induced anosmia or hyposmia in response to inflammation [46,47].

This study describes the effect of environmental housing conditions on the murine VNO and its intraspecific chemical communication capabilities. It also suggests a possible link between these changes and the reduction of animal welfare. Animal density is a well-established cause of reduced welfare due to high contact between individuals [48] but also due to the increase in irritant gas and dust emitted from liter degradation, which has been proven to be responsible for respiratory tract inflammation [42,49], and suggested to be a potential cause of vomeronasalitis [13]. The presence of this alteration was associated with behavioral modifications and an increase in aggression in the affected animals [13]. This finding confirms the essential role of this organ in intraspecific communication and, thus, its role as a cofactor in the general welfare of these animals. This study highlights the importance of animal housing environments and demonstrates the negative impact of a confined environment and the presence of ammonia on vomeronasal organ physiology. It provides new insights concerning the importance of animal density in farming and its impact on chemical communication, animal behavior, and thus on animal welfare.

## 5. Conclusions

This work showed that the mouse vomeronasal organ is significantly impacted by a confined environment, with an increase in degenerative and inflammatory indicators in most VNO structures and a modification of neuronal protein expression in the vomeronasal sensory epithelium. The presence of ammonia appeared to increase the severity of the changes in the VNO conditions, but further analyses are needed to validate the effect of this irritant gas and other litter components. Considering the high density and the confined environment in which farm animals live, exposure to these contaminants appears to be the most important cause of VNO alterations in these species, particularly in pigs, which have already been reported to be affected by vomeronasalitis with a consequent increase in aggressive behaviors in the group [13,24]. In conclusion, although further studies are needed to confirm our findings, this study revealed the possible causes of VNO changes in animals and improved the understanding of how the living environment can influence the chemical communication, behavior, and welfare of farm animals.

## Figures and Tables

**Figure 1 animals-13-01902-f001:**
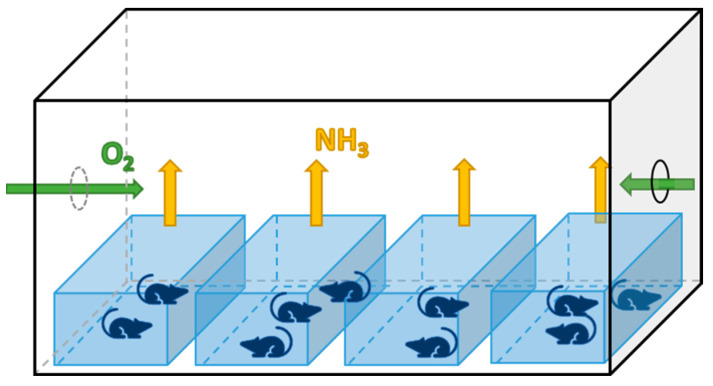
Schematic of the plexiglass box setup used to obtain the confined conditions. O_2_ = Dioxygen; NH_3_ = Ammonia.

**Figure 2 animals-13-01902-f002:**
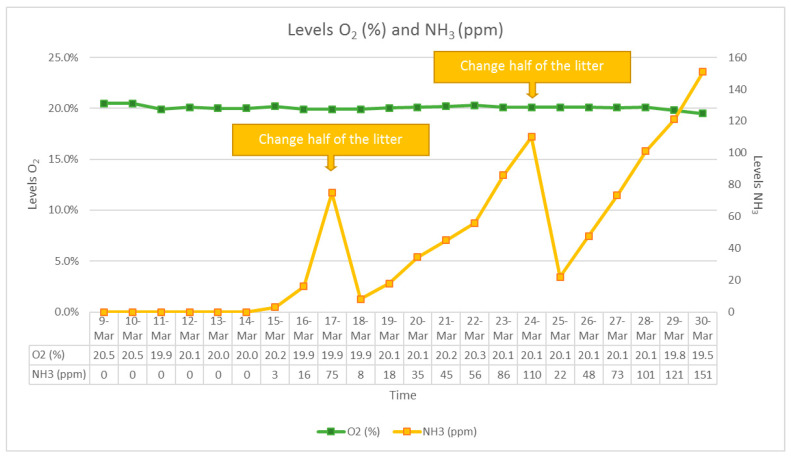
Daily levels of dioxygen (in green), and ammonia (in yellow), in the device during the experiment for the group exposed to ammonia.

**Figure 3 animals-13-01902-f003:**
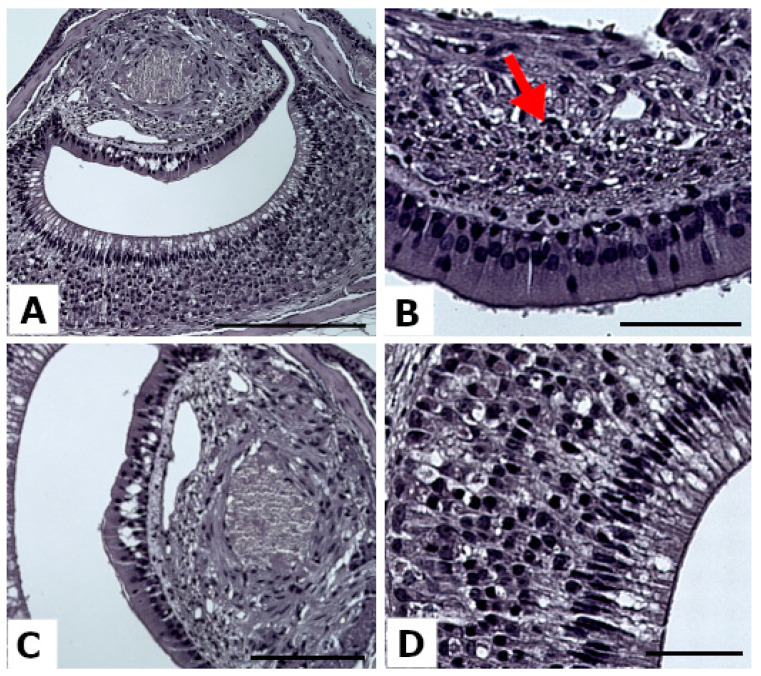
Representative images of a score “2” in the tested parameters after hematoxylin and eosin staining. (**A**) Mouse VNO alterations, Scale bar = 200 µm. (**B**) Soft tissue alteration with a lymphocyte infiltration (red arrow). Scale bar = 50 µm; (**C**) NSE alteration, scale bar = 100 µm; (**D**) VNSE alteration, scale bar = 50 µm.

**Figure 4 animals-13-01902-f004:**
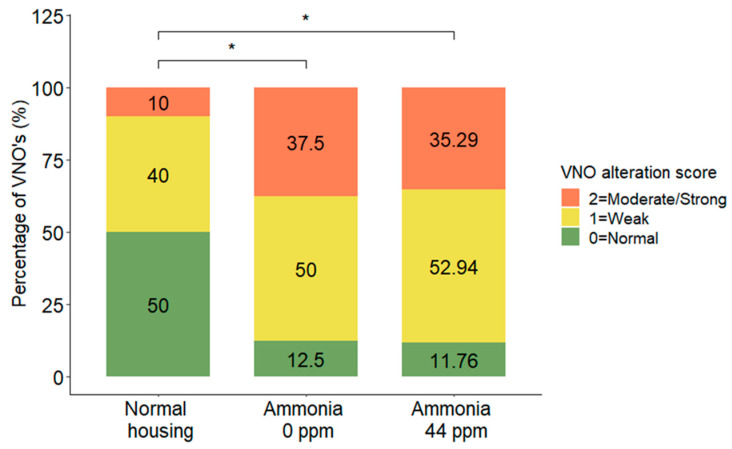
Distribution (%) of VNO alteration scores in the VNOs of mice in normal housing conditions, confined conditions, or confined conditions with natural ammonia exposure. Data are shown as the percentage of VNOs for each score in the three different groups, * *p* ≤ 0.05.

**Figure 5 animals-13-01902-f005:**
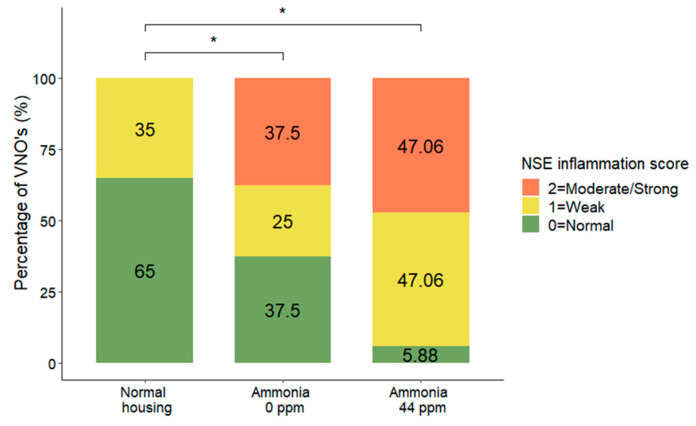
Distribution (%) of nonsensory epithelium inflammation scores comparing mouse NSEs after normal housing conditions, confined conditions, or confined conditions with natural ammonia exposure. Data are shown as the percentage of VNOs for each score, * *p* ≤ 0.05.

**Figure 6 animals-13-01902-f006:**
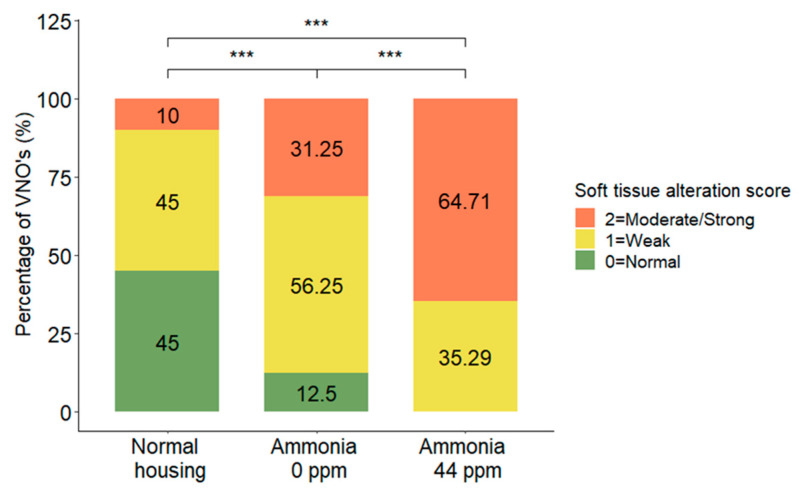
Distribution (%) of VNO soft tissue alteration scores of mice housed in normal housing conditions, confined conditions, or confined conditions with natural ammonia exposure. Data are shown as the percentage of VNOs for each score, *** *p* ≤ 0.001.

**Figure 7 animals-13-01902-f007:**
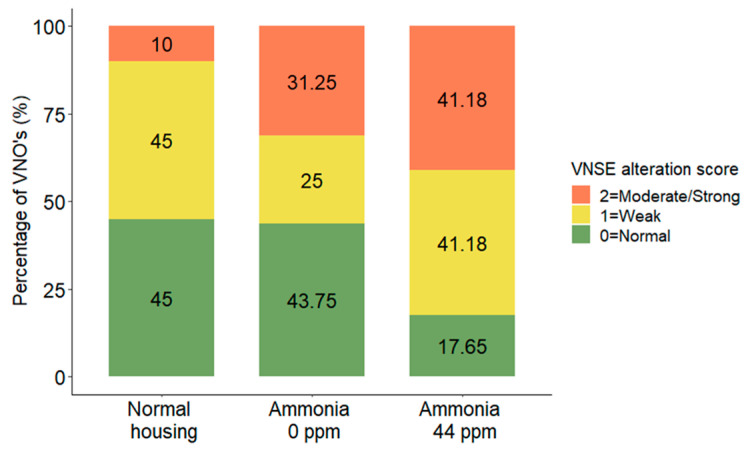
Distribution (%) of vomeronasal sensorial epithelium alteration scores comparing the VNSEs of mice housed in normal conditions, confined conditions, or confined conditions with natural ammonia exposure. Data are shown as the percentage of VNOs for each score.

**Figure 8 animals-13-01902-f008:**
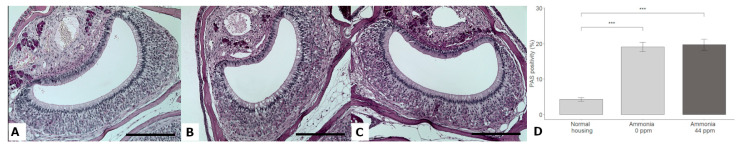
Effects of environmental living conditions on glycogen accumulation. PAS staining was used to stain glycogen accumulation in the VNSEs of mice housed in normal conditions (**A**), confined conditions (**B**), or confined conditions with natural ammonia exposure (**C**). Data are shown (**D**) as the mean ± SD, *** *p* ≤ 0.001. Objective × 20, Scale bar = 200 µm.

**Figure 9 animals-13-01902-f009:**
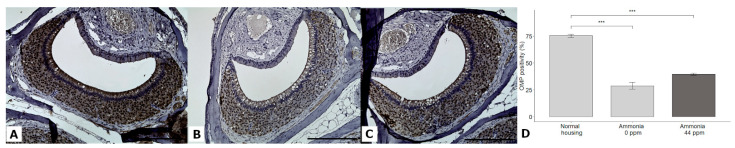
Immunohistochemical staining of the OMP protein in the VNSEs of mice housed in normal conditions (**A**), confined conditions (**B**), or confined conditions with natural ammonia exposure (**C**). Data are shown (**D**) as the mean ± SD, *** *p* ≤ 0.001. Objective × 20, Scale bar = 200 µm.

**Figure 10 animals-13-01902-f010:**
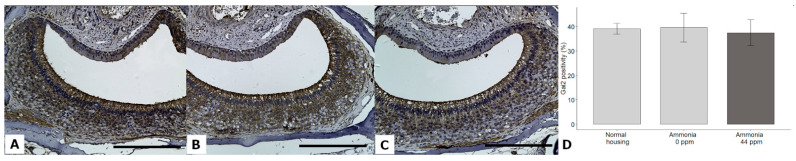
Immunohistochemical staining of the Gαi2 protein in the VNSEs of mice housed in normal conditions (**A**), confined conditions (**B**), or confined conditions with natural ammonia exposure (**C**). Data are shown (**D**) as the mean ± SD. Objective × 20, Scale bar = 200 µm.

**Figure 11 animals-13-01902-f011:**
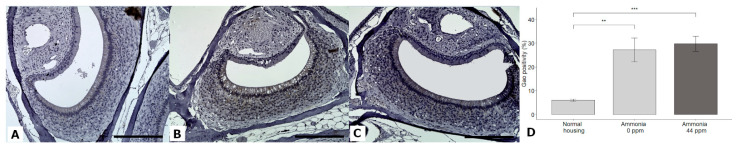
Immunohistochemical staining of the Gαo protein in the VNSEs of mice housed in normal conditions (**A**), confined conditions (**B**), or confined conditions with natural ammonia exposure (**C**). Data are shown (**D**) as the mean ± SD, ** *p* ≤ 0.01; *** *p* ≤ 0.001. Objective × 20, Scale bar = 200 µm.

**Table 1 animals-13-01902-t001:** Distribution of the histochemical and immunohistochemical parameters according to the environmental air conditions. Data are shown as the percentage of VNOs for each score.

Qualitative Variables						
Variable	N (%) Normal Housing (N tot = 20)	N (%) Ammonia 0 ppm (N tot = 16)	N (%) Ammonia 44 ppm (N tot = 17)	Model Used	Global Results	Multiple Comparisons
**VNO alteration**				MOLR	χ^2^ = 5.95, DF = 2, *p* = 0.0311	Normal < w NH_3_ * Normal < w/o NH_3_ *
0 = Normal	10 (71.4)	2 (14.3)	2 (14.3)			
1 = Weak	8 (32.0)	8 (32.0)	9 (36.0)			
2 = Moderate/strong	2 (14.3)	6 (42.9)	6 (42.9)			
**NSE inflammation**				MOLR	χ^2^ = 12.04, DF = 2, *p* = 0.0024	Normal < w NH_3_ * Normal < w/o NH_3_ *
0 = Normal	13 (65.0)	6 (30.0)	1 (5.0)			
1 = Weak	7 (36.8)	4 (21.1)	8 (42.1)			
2 = Moderate/strong	0 (0.0)	6 (42.9)	8 (57.1)			
**Soft tissue alteration**				MOLR	χ^2^ = 5.91, DF = 2, *p* = 0.0480	Normal < w NH_3_ *** Normal < w/o NH_3_ *** w NH_3_ > w/o NH_3_ ***
0 = Normal	9 (81.8)	2 (18.2)	0 (0.0)			
1 = Weak	9 (37.5)	9 (37.5)	6 (25.0)			
2 = Moderate/strong	2 (11.1)	5 (27.8)	11 (61.1)			
**VNSE inflammation**				MOLR	χ^2^ = 2.98, DF = 2, *p* = 0.2251	NA
0 = Normal	9 (47.4)	7 (36.8)	3 (15.8)			
1 = Weak	9 (45.0)	4 (20.0)	7 (35.0)			
2 = Moderate/strong	2 (14.3)	5 (35.7)	7 (50.0)			
**Quantitative variables**						
**Variable**	Mean ± SD Normal housing	Mean ± SD Ammonia 0 ppm	Mean ± SD Ammonia 44 ppm	Model used	Global results	Multiple comparisons
**PAS positivity (%)**	4.24 ± 1.95	19.05 ± 5.01	19.70 ± 6.45	GLMM	χ^2^ = 47.92, DF = 2, *p* < 0.0001	Normal < w NH_3_ *** Normal < w/o NH_3_ ***
**OMP positivity (%)**	75.27 ± 7.07	28.77 ± 12.83	39.56 ± 4.22	GLMM	χ^2^ = 138.48, DF = 2, *p* < 0.0001	Normal > w NH_3_ *** Normal > w/o NH_3_ ***
**Gαi2 positivity (%)**	39.09 ± 9.91	39.56 ± 24.48	37.47 ± 22.49	GLMM	χ^2^ = 1.88, DF = 2, *p* = 0.3892	NA
**Gαo positivity (%)**	6.03 ± 1.95	27.22 ± 21.32	29.82 ± 13.49	GLMM	χ^2^ = 8.95, DF = 2, *p* = 0.0114	Normal < w NH_3_ Normal < w/o NH_3_

MOLR = mixed ordinal logistic regressions; GLMM = general linear mixed model; χ^2^ = chi-squared test; DF = degrees of freedom; Normal = normal conditions; w/o NH_3_ = confined conditions without NH_3_; w NH_3_ = confined conditions in the presence of NH_3_; * = *p* ≤ 0.05; *** = *p* ≤ 0.001.

## Data Availability

Not applicable.

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
