# Peer review of "Does the Environmental Air Impact the Condition of the Vomeronasal Organ? A Mouse Model for Intensive Farming"

_animals, 2023, doi:10.3390/ani13121902_

Round 1

Reviewer 1 Report

    The starting point of this work is rationale.

From the experimental design, if you want to examine the effects of environmental anmonia (The most important pollutant in mouse houses) on VNO structure ,  is it necessary to compare with normally housed mice?

     There are a lot of structural differences of VNO between confined conditions (with or without ammonia supply) and normally caging condition . On the contrary, there is little difference betwee nammonia supply group and ammonia-free group, implying environmental ammonia has no effects on VNO structure. Instead, youshould  make a conclusion with no efefcts of air-borne ammonia has no effects on VNO.

    I wonder whether the concentraion of supplied ammonia is sufficent or not.  In future work,  I suggested that you can try to test if the behavioral responses of mice to sex pheromones are changed, and even exmiane the expression of vomeralnasal receptors via RNA-seq. 

Reviewer 2 Report

Dear authors

This is an interesting topic. I appreciated your article. I think it is very interesting and well done. The methods imployed  histopathological and immunoistochemical analyses, are sufficiently reported, the experimental model is appropriated. Interpretation of the results is adequate. Conclusions are supported by the results.

Author Response

Thank you very much for your comments and your appreciation of our work. 

Reviewer 3 Report

The manuscript written by Mechin et al. revealed the effect of poor environment on the vomeronasal organ. Research from this perspective is essential and arguably invaluable for advancing animal welfare knowledge. The effects to the vomeronasal organ at concentrations likely to occur in everyday life in animal husbandry are surprising. There are two things I would like to confirm, but I think both are minor.

Main comment 1.

Discussion, Paragraph 4, Line 309-: Authors discussed that the reason why there was no significant difference was that the concentration of ammonia was not high enough. This logic is extremely dangerous from an animal welfare perspective, because readers may consider that it is recommended to reproduce at higher concentrations during confirmatory experiments. Actually, 44 ppm of ammonia is near the acceptable limit for animal experiments. I recommend a description like “The difference is not significant, but the impact would be stronger on farms with poorer environmental conditions” here.

Main comment 2.

The present immunostaining results are quantified by the number of positive pixels, and their significant differences are verified. Should this change be regarded as a change in the number of positive cells? Or did the expression levels in a single cell change?

Specifically: Is it correct that a hostile environment decreases the number of mature receptor cells expressing OMPs, but increases the number of receptor cells expressing V2Rs?

Minor comments.

1. Line 102: The numeral “2” should be subscripted.

2. Figure 2: A mean NH3 level (44 ppm) should not be contained in this graph. Maybe this parameter was total ammonia concentration divided by the number of experimental days, and no need to indicate by date.

3. Figure 3: Was these sections really stained with eosin? It is completely different from the correct staining manner of hematoxylin-eosin stain.

4. Figure 8. Line 250: One of “(B)” should be “(C)”.

Reviewer 4 Report

The authors address the effects of gaseous litter emissions on changes in the health status of the vomeronasal organ of mice, used here as a model organism for indoor farming conditions.

Three groups/conditions of each 10 mice were analyzed. The main finding is that harmful volatiles emitted by decomposing biomass alter vomeronasal health/function and may affect social behavior.

Overall, the study is well-written and organized.

My concerns and questions are as listed below.

Major concern

Statistical table 1: The 10 mice used in each group were artificially expanded to 20 VNOs. Since the left and right VNO belong to the same mouse (dependency), they cannot be treated as independent or individual data points. It is highly unlikely that the left VNO would show a different result to the right VNO in a single mouse. I suggest recalculating for statistical correctness.

Fig. 3: It would be helpful for the reader to see representative images of all 3 categories. Fig.3 shows only level 2 changes.

Minor concerns

VNO tissue sections: How many VNO tissue sections were analyzed and scored for changes in each mouse? Does the repertoire of sections cover the anterior, middle, and posterior extent of the VNO?

Gender: Regarding gender, although homogeneity was observed, how many males and females were analyzed? Please add numbers.

Selected area (lines 168- 169) “…was expressed as the percentage of positive pixels in the selected area, corresponding to the VNSE, as previously reported (23).” 

à How is the selected area defined? No further information is found when checking reference (23). Please describe the defined area. Similarly, lines 141-143 (selected surface). Was the entire extent of the VNO been analyzed or only a defined square [xy µm2]?

Plexiglas box: What is the diameter of the lateral holes that allow gas exchange with the external environment? Have the two plexiglas-enclosed groups been in the same room with the control mice (plus other mice from other studies)? This is important to know because the two confined groups show differences compared with the normal housed control mice but very few differences between them (with/without ammonia). It seems that the effect of ammonia is rather insignificant, while the plexiglas housing may well have an effect compared to the normally housed mice.

Images: It is difficult to see changes in the images shown. Please use arrows to indicate changes and include inset magnifications of affected areas.  

Fig. 8D: The labelling is too small, please increase the font size.

Lines 130-131: “The presence of abnormal amounts of lymphocytes was then classified as 0 (healthy), 1 (few lymphocytes), and 2 (strong lymphocyte infiltration).”  It would be helpful to show a picture with arrows showing the infiltration observed.     

Wording

Line 47/48: “vomeronasal type 1 /type 2 receptors”: the authors probably refer to the vomeronasal neurons and not to the receptors à Please add “cells” or “neurons” throughout the manuscript. The same applies to V1R and V2R.

Line 112: delete humanely or replace with euthanized with … according to EU animal welfare standards.

Line 113: “sodic” ß à “sodium”

General comment: Yes, most likely the amount of ammonia was not high enough to cause adequate changes in the VNO. In this context, it would have been quite interesting to see what has happened in the main olfactory epithelium. These data would add a bit more substance to the manuscript.

General question

Did the authors observe any behavioural changes following ammonia exposure?

English is OK
